# Photobiomodulation for Parkinson’s Disease in Animal Models: A Systematic Review

**DOI:** 10.3390/biom10040610

**Published:** 2020-04-15

**Authors:** Farzad Salehpour, Michael R Hamblin

**Affiliations:** 1College for Light Medicine and Photobiomodulation, D-82319 Starnberg, Germany; farzadsalehpour1988@gmail.com; 2Niraxx Light Therapeutics, Inc., Irvine, CA 92617, USA; 3ProNeuroLIGHT LLC, Phoenix, AZ 85083, USA; 4Wellman Center for Photomedicine, Massachusetts General Hospital, Harvard Medical School, Boston, MA 02114, USA; 5Laser Research Centre, Faculty of Health Science, University of Johannesburg, 2028 Doornfontein, South Africa

**Keywords:** Parkinson’s disease, animal models, photobiomodulation, low-level laser therapy, transcranial, abscopal, parameters

## Abstract

Photobiomodulation (PBM) might be an effective treatment for Parkinson’s disease (PD) in human patients. PBM of the brain uses red or near infrared light delivered from a laser or an LED at relatively low power densities, onto the head (or other body parts) to stimulate the brain and prevent degeneration of neurons. PD is a progressive neurodegenerative disease involving the loss of dopamine-producing neurons in the substantia nigra deep within the brain. PD is a movement disorder that also shows various other symptoms affecting the brain and other organs. Treatment involves dopamine replacement therapy or electrical deep brain stimulation. The present systematic review covers reports describing the use of PBM to treat laboratory animal models of PD, in an attempt to draw conclusions about the best choice of parameters and irradiation techniques. There have already been clinical trials of PBM reported in patients, and more are expected in the coming years. PBM is particularly attractive as it is a non-pharmacological treatment, without any major adverse effects (and very few minor ones).

## 1. Introduction

Parkinson’s disease (PD) is a multifactorial and multisystem disease, characterized by the loss of the dopamine producing neuronal cells of the substantia nigra pars compacta (SNc) in the brain [1,2]. The lack of dopamine primarily affects the motor function, but there are many other signs and symptoms that affect mood, cognition, digestive system, sense of smell, etc. The motor symptoms include bradykinesia, muscular rigidity, tremor at rest, and postural instability. The dopamine producing neurons die off, and one somewhat controversial theory to explain this is the accumulation of Lewy bodies containing aggregated α-synuclein inside the cells. The causes of PD are not completely understood. Only about 15% of PD patients are likely to have a genetic cause, among which mutations in leucine-rich repeat kinase 2 (LRRK2), GBA1 (glucocerebrosidase), and SNCA (α-synuclein) are the most common [3]. The environmental causes are complex, but recent evidence has implicated mitochondrial dysfunction [4] and changes in the gut microbiome [5]. Over 1 million individuals in the US suffer from PD and the annual financial burden is estimated to be $52 billion [6]. The accepted treatment is replacement of the lost dopamine using Levodopa, which helps the motor symptoms but does not modify the course of the disease [7]. Monoamine oxidase-B inhibitors and dopamine agonists might be used later in the course of the disease. Deep brain stimulation (DBS) using an electrode implanted into the subthalamic nucleus and other brain regions has also shown promising results [8].

Photobiomodulation (PBM) involves the use of low-powered red and near-infrared (NIR) light from a laser or light-emitting diode (LED) to stimulate, heal, and regenerate damaged or dying tissues [9]. PBM was previously known as low-level laser (light) therapy (LLLT) [10]. PBM was discovered by Endre Mester soon after the first ruby laser was discovered by Ted Maiman in 1960 [11]. For many years, it was thought that a coherent laser beam was necessary for effective PBM [12], but now it is appreciated that in many situations, LEDs might be a better choice [13]. The mechanism of action primarily involves absorption of the light through the mitochondria, leading to an increased membrane potential, electron transport, oxygen consumption, and ATP synthesis [9]. Since the brain is heavily dependent on mitochondrial activity, it is not surprising that PBM has been extensively tested to treat various brain disorders [14]. Many signaling pathways are activated by PBM, including those mediated by reactive oxygen species (ROS), leading to the up-regulation of anti-oxidant defenses [15]. Anti-apoptotic and pro-survival signaling is also activated [16]. Moreover the ability to switch mitochondrial respiration from glycolysis towards oxidative phosphorylation has two other important effects. First, stem cells are mobilized from their hypoxic niche and can migrate towards sites of injury where they can repair the damage [17]. Second, the mitochondrial alteration can switch the macrophage and microglial phenotype from the pro-inflammatory M1 state, to the anti-inflammatory and phagocytic M2 state [18]. In the brain, neurotrophic factors (such as brain-derived neurotrophic factor [BDNF]) are up-regulated [19], adult hippocampal neurogenesis is stimulated [20], and synaptogenesis and neuroplasticity is encouraged [19].

These latter effects can be thought of as “helping the brain to repair itself”, and suggest that PBM can be useful for many traumatic brain disorders, such as stroke [21] and traumatic brain injury [22], as well as neurodegenerative brain disorders like Alzheimer’s disease [23] and PD [24]. One question that is often asked about PBM for the brain, is how important is it to apply the light to the head and for the photons to actually penetrate into the brain tissue, or else how important is it for the light to be absorbed by the circulating blood or bone marrow? The latter pathways might explain the systemic or abscopal effects of PBM, which have been reported by many authors [25]. The recent discovery of respiratory-competent cell free mitochondria that are circulating in the blood of normal individuals [26] might offer an explanation for how the beneficial effects of light that is incident on the body can be transmitted to distant organs including the brain. Calculation or measurement of the fraction of photons that are incident on the scalp and which penetrate the cortex, and especially into deeper brain structures (such as the SNc), is not particularly encouraging [27], suggesting that for PD, the abscopal effect, or the application of light to the abdomen, to affect the gut microbiome (“photobiomics” [28]), might be important.

The goal of the present paper was to undertake a systematic review of published studies, which have examined the use of PBM therapy to treat PD in animal models, to see if any conclusions about the parameters and methods can be drawn.

## 2. Materials and Methods

### 2.1. Search Strategy

The primary search was conducted from 1990 to November 2019. Bibliographic databases (i.e., MEDLINE through PubMed, SCOPUS, Web of Science, EMBASE and Cochrane Library) were searched electronically for studies on the neuroprotective effects of PBM on animal models of PD, through the keywords “photobiomodulation”, “low-level light therapy”, “low-level laser therapy”, “near-infrared light”, “red light”, “Parkinson’s disease”, and “Parkinsonism”. Two independent investigators screened the title, abstract and the full text of the articles and judged the searched materials against the inclusion and exclusion criteria. The search was limited to the original studies performed in animals and to publications written in English. Therefore, ex vivo, in vitro or clinical original articles, as well as review articles were not included.

### 2.2. Inclusion and Exclusion Criteria

We included all in vivo studies reporting the effects of PBM, as opposed to vehicles, on the behavioral and molecular outcomes in PD models. Studies conducting PBM via transcranial, intracranial, systemic irradiation (remotely or laser acupuncture irradiations) as well as whole-body irradiation approaches in PD models were included. Studies performed on ex vivo or in vitro (primary cultures or cell line), as well as clinical trials, were excluded. Additionally, studies conducted on intact (healthy) animals were excluded from our review. Moreover, non-English language publications and studies involving NIR spectroscopy and conference papers were excluded.

### 2.3. Data Extraction

The author, publication year, animals and species, number of animals in each experimental group, gender and age, type of PD model, light source/wavelength, output power, irradiance (power density), irradiation time, fluence (energy density) or energy (dose), total fluence or dose, irradiation approach/site, number of treatment sessions, and outcome(s) were extracted. However, the time of outcome evaluation was not extracted from the studies.

## 3. Results

The initial systematic search of the mentioned databases identified 354 articles, of which 28 studies met the inclusion criteria (Figure 1). Twenty-two articles reported experiments in rodents, five articles reported studies in primates (macaque monkey, *Macaca fascicularis*), and one study was conducted in a Pink1 mutant PD model. Of the twenty-two studies on rodents, sixteen studies assessed the effects of PBM in mice, of which thirteen were on the albino BALB/c strain and three were on the C57BL/6 strain. Additionally, six rodent studies were performed on rats, of which five were on the Sprague–Dawley strain and one was on the albino Wistar strain. It should be noted that in one study, more than one experiment was conducted using three different animal species of BALB/c mice, Wistar rats, and macaque monkeys; and also in one study, two different types of irradiation methods, transcranial or remote-tissue were performed; in these cases, each experiment was regarded as a separate study and was included in the systematic review.

Animal models of PD were induced using injections of methyl-4-phenyl-1,2,3,6-tetrahydropyridine (MPTP) in mice or primates. Other models used 6-hydroxydopamine (6OHDA) in rats, and rotenone in Drosophila Pink1 mutants. In the context of molecular and biochemical assessments, the possible neuroprotective effects of PBM were evaluated in various brain regions, including the SNc, subthalamic nucleus (STN), striatum, zona incerta (ZI), zona incerta-hypothalamus (ZI-Hyp), caudate putamen (CPu) and periaqueductal grey matter (PaG).

In fifteen studies, laser or LED light was delivered to the head of the animal in a transcranial approach. On the other hand, nine studies used an intracranial irradiation approach via implantation of an optical fiber connected to a light source into the region of interest inside the brain. In addition, four studies performed systemic PBM using remote-tissue irradiation (abscopal effect) or laser acupuncture methods. Whole-body PBM was carried out in one study of Pink1 Drosophila mutant PD model. Eighteen studies applied LED-based devices, while eleven studies used lasers as light sources. Twenty six studies performed PBM with red/far-red wavelengths (627 nm [one study], 630 nm [one study], 670 nm [twenty one studies], and 675 nm [two studies]), whereas, four studies used NIR light (808 nm) and only in one study blue light (405 nm) was delivered via an acupuncture point. The operation mode of light sources in all studies was a continuous wave (CW). Other physical treatment parameters, such as output power, irradiance, irradiation time, fluence, total delivered dose, numbers and duration of treatment sessions are summarized in Table 1.

## 4. Discussion

The evidence that has been presented in this systematic review does suggest that PBM (and in particular transcranial PBM) is an effective method to treat animal models of PD. The discovery of the toxic effects of MPTP, which is an impurity found in recreational drugs consumed by individuals in San Francisco in 1982, for the first time allowed the creation of laboratory animal models of PD [52]. Besides MPTP, other compounds have been used to produce PD-like models [53], including 6-hydroxydopamine (6-OHDA) paraquat, rotenone, and Maneb (a polymeric Mn complex of ethylene bis (dithiocarbamate). The mechanism of action of these compounds usually involves metabolism into intermediates that can undergo redox cycling and thereby damage the mitochondria, and in particular Complex 1. There have also been genetic models of PD involving mutations to genes such as α-synuclein, Parkin (an ubiquitin E3 ligase), PINK1 (PTEN-induced putative kinase 1), and LRRK2 (leucine-rich repeat kinase 2). Although the animal models of PD do not completely mimic the human disease, they have been useful for studying the pathophysiology of PD, and for testing the effectiveness of novel treatments, including DBS and PBM. It is expected that further animal studies will use PBM in genetically engineered models of PD rather than toxin-induced models, because these are now considered to be more representative of the human disease.

Although most of animal studies have used red light (670 nm, 675 nm or 630 nm), this does not necessarily mean that red wavelengths are better than NIR wavelengths (810 nm). This preponderance might simply reflect the wider use of red LEDs in ophthalmology and wound healing. The power density levels employed were generally between 20–50 mW/cm^2^, but occasionally lower or higher values were employed. Moderate illumination times (minutes) generally provided fluences in the range of 10–60 J/cm^2^ on the scalp. The intracranial fibers that were implanted into the brain delivered fairly low powers (up to 14 mW), but when the illumination was continued for several days, the total energy density delivered could be quite large. It should be noted that the regions of the brain where optical fibers are implanted are different from the regions where electrodes are implanted in the DBS procedure. In DBS, electrodes are usually implanted into the globus pallidus internus to improve the motor function [54] or into the subthalamic nucleus [55] or the caudal zona incerta to improve tremor [56]. The optical fibers in PD animal models have been implanted into the mid-brain, with the goal of delivering the light as close as possible to the SNc, to preserve the dopamine producing neurons.

Pulsing is an interesting parameter for brain PBM therapy, as it has been found that pulsing the light at certain frequencies is more effective than CW light [57]. The two most popular frequencies are 10 Hz (the so-called alpha rhythm) and 40 Hz (the so-called gamma rhythm). The idea is that these frequencies can resonate with intrinsic brain rhythms, and therefore, can improve brain function to a greater extent than CW light [57]. The repetition regimens that have been used for treating the animal models of PD range from a few times per day to every few days, for periods that could be as long as 4 weeks. As PD in humans is a chronic degenerative disease, it is expected that PBM therapy would need to be continued for the foreseeable future.

The encouraging results that have been obtained in the animal studies reviewed above have led to the initiation of clinical studies of PBM therapy for PD patients. Hamilton and colleagues described the construction of “light buckets” lined with LEDs (670, 810 and 850 nm) to treat patients with PD [58] (Figure 2). These devices delivered a power density of 10 mW/cm^2^ to the entire head, and in addition an intranasal device with a power of 4 mW/cm^2^ was employed. Patients were treated twice a day (1800 J per session) for 30 days. The initial symptoms of tremor, akinesia, gait, difficulty in swallowing and speech, poor facial animation, and reduced fine motor skills, loss of the sense of smell, and impaired social confidence were all improved in ~75% of the subjects, while ~25% remained the same and none got worse. The improvements were still maintained over an extended period (up to 24 months). Santos et al. conducted a randomized controlled trial in Parkinson’s patients using a CW 670 nm LED array (WARP 10) over 10 cm^2^, on 6 sites on both temples at 60 mW/cm^2^, delivering 6 J/cm^2^ and a total energy of 2160 J [59]. A total of 18 sessions were given over 9 weeks leading to clinical improvements.

Additional clinical trials are in progress that, in addition to applying light to the head also apply light to the abdomen, with the goal of improving the gut microbiome. The results are eagerly awaited.

## Figures and Tables

**Figure 1 biomolecules-10-00610-f001:**
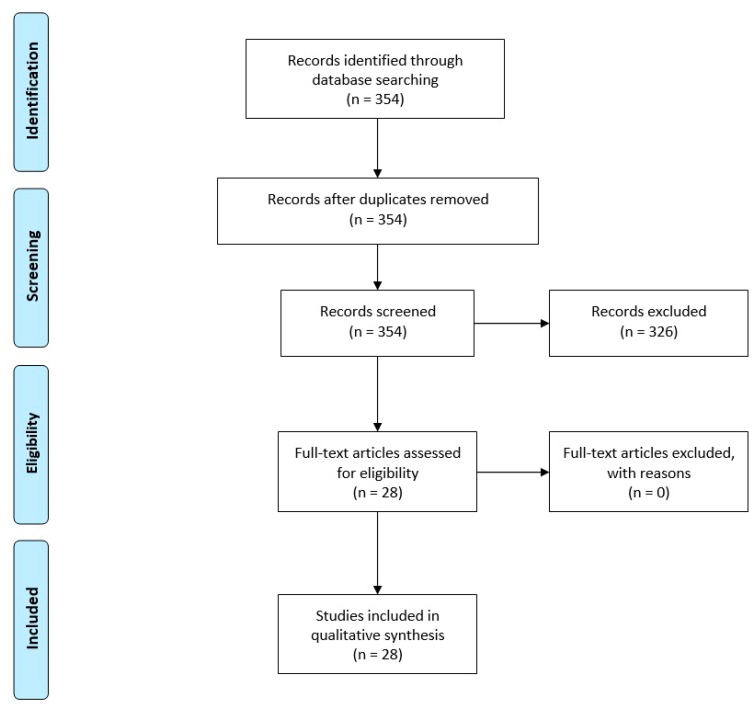
Systematic review flow chart for the inclusion of eligible studies.

**Figure 2 biomolecules-10-00610-f002:**
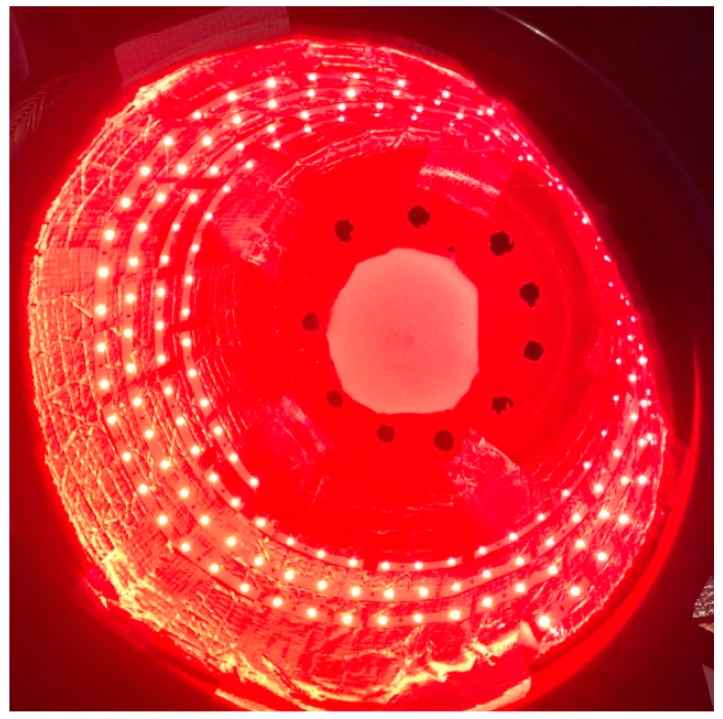
Photograph of the “light bucket” described by Hamilton et al. [24].

**Table 1 biomolecules-10-00610-t001:** Summary of Studies on the Effects of Photobiomodulation Therapy in Animal Models of Parkinson’s Disease.

Study/Year	Animal/Species (n)	Gender/Age	PD Model	Light Source	Output Power	Irradiance	Irradiation Time per Session	Fluence or Dose per Session	Total Fluence or Dose	Irradiation Approach/Sites	Number of Treatment Sessions	Outcomes
Shaw et al., (2010) [29]	Mouse Albino BALB/c (n _Saline_ = 20) (n _Saline + PBM_ = 20) (n _MPTP_ = 20) (n _MPTP + PBM_ = 20)	Male 8 weeks old	MPTP Mild: 50 mg/kg per mouse Strong: 100 mg/kg per mouse	LED, 670 nm	NR	40 mW/cm^2^ (at scalp)	90 s	3.6 J/cm^2^ (at scalp)	14.4 J/cm^2^ (at scalp)	Transcranial Holding probe at 1 cm from the head	4 simultaneous irradiations over 30 h	Increased TH^+^ terminals in the caudate-putamen complex; no effect on the overall volume of the SNc and ZI-Hyp; increased TH^+^ cells in the SNc and ZI-Hyp regions; no effect on the morphology of TH^+^ cells in both the SNc and ZI-Hyp; increased number of TH^+^ cells in the SNc (in both 50 and 100 mg/kg MPTP doses); no effect on the number of TH^+^ cells in the ZI-Hyp (in 50 and 100 mg/kg MPTP doses)
Peoples et al., (2012) [30]	Mouse Albino BALB/c (n _Saline_ = 20) (n _Saline + PBM_ = 20) (n _MPTP_ = 20) (n _MPTP+PBM_ = 20)	Male 8 weeks old	MPTP: 200 mg/kg per mouse	LED, 670 nm	NR	40 mW/cm^2^ (at scalp)	90 s	3.6 J/cm^2^ (at scalp)	Simultaneous group: 36 J/cm^2^ (at scalp) Post-treatment group: 36 J/cm^2^ (at scalp)	TranscranialHolding probe at 1–2 cm from the head	Simultaneous group: 10 irradiations over 5 weeks Post-treatment group: 10 irradiations over 3 weeks	For both simultaneous and post-treatment series: increased TH^+^ cell number in the SNc, but not in the PaG and ZI-Hyp regions
Shaw et al., (2012) [31]	Mouse Albino BALB/c (n _Saline_ = 24) (n _Saline + PBM_ = 24) (n _MPTP_ = 24) (n _MPTP + PBM_ = 24)	Male 8 weeks old	MPTP Acute: 100 mg/kg per mouse Chronic: 200 mg/kg per mouse	LED, 670 nm	NR	40 mW/cm^2^ (at scalp)	90 s	3.6 J/cm^2^ (at scalp)	Acute regimen: 14.4 J/cm^2^ (at scalp) Chronic regimen: 36 J/cm^2^ (at scalp)	TranscranialHolding probe at 1–2 cm from the head	Acute regimen: 4 simultaneous irradiations over 30 h Chronic regimen: 10 simultaneous irradiations over 5 weeks	For acute regimen: decreased Fos^+^ cell number in the STN and ZI regions in group with six-day survival period For chronic regimen: decreased Fos^+^ cell number in the STN and ZI regions
Peoples et al., (2012) [32]	Mouse Albino BALB/c (n _Saline_ = 21) (n _Saline + PBM_ = 19) (n _MPTP_ = 22) (n _MPTP + PBM_ = 18)	Male 8 weeks old	MPTP Acute: 100 mg/kg per mouse Chronic: 200 mg/kg per mouse	LED, 670 nm	NR	40 mW/cm^2^ (at scalp)	90 s	3.6 J/cm^2^ (at scalp)	Simultaneous acute group: 14.4 J/cm^2^ (at scalp) Simultaneous chronic group: 36 J/cm^2^ (at scalp) Post-treatment acute group: 14.4 J/cm^2^ (at scalp) Post-treatment chronicgroup: 36 J/cm^2^ (at scalp)	TranscranialJust above the mouse head and in full view of the eyes	Simultaneous group: 4 irradiations over 30 h (acute regimen) or 10 irradiations over 5 weeks (chronic regimen) Post-treatment group: 4 irradiations over 2 days (acute regimen) or 10 irradiations over 3 weeks (chronic regimen)	For all group and regimens: no effect on the retinal areas For all groups except simultaneous group with acute regimen: increased TH^+^ cell number in the retina
Moro et al., (2013) [33]	Mouse Albino BALB/c: (n _Saline_ = 10) (n _Saline + PBM_ = 10) (n _MPTP_ = 10) (n _MPTP + PBM_ = 10) Black C57BL/6: (n _Saline_ = 10) (n _Saline + PBM_ = 10) (n _MPTP_ = 10) (n _MPTP + PBM_ = 10)	Male 8–10 weeks old	MPTP: 50 mg/kg per mouse	LED, 670 nm	NR	40 mW/cm^2^ (at scalp)	90 s	3.6 J/cm^2^ (at scalp)	14.4 J/cm^2^ (at scalp)	TranscranialHolding probe at 1–2 cm from the head	4 simultaneous irradiations over 30 h	For Albino BALB/c mice: increased TH^+^ cell number in the SNc; improved locomotor activities via increase of velocity and high mobility, and decrease of immobility For C57BL/6 mice: no effect on the TH^+^ cell number in the SNc; no effect on the locomotor activities
Purushothuman et al., (2013) [34]	Mouse K3 transgenic model (n _WT_ = 5) (n _K3_ = 5) (n _K3 + PBM_ = 5)	NR 5 months old	K369I tau transgenic model	LED 670 nm	NR	40 mW/cm^2^ (at scalp)	90 s	4 J/cm^2^ (at scalp)	80 J/cm^2^ (at scalp)	TranscranialHolding probe at 1–2 cm from the head	20 irradiations over 4 weeks	Decreased markers of oxidative stress, over expression of hyperphosphorylated tau, and increased TH^+^ cell number in the SNc
Vos et al., (2013) [35]	*Drosophila Pink1* null mutants	NA	Rotenone (250 μM)	Laser, 808 nm	NR	25 mW/cm2	100 s	2.5 J/cm^2^	2.5 J/cm^2^	Whole-body	One session (single dose)	Improved CCO-dependent oxygen consumption and ATP production; rescued major systemic and mitochondrial defects
Wattanathorn and Sutalangka, (2014) [36]	Rat Albino Wistar (n _Control_ = 12) (n _6OHDA_ = 12) (n _6OHDA + Sham PBM_ = 12) (n _6OHDA + Sham PBM_ = 12)	Male 8 weeks old	6OHDA (6 μg per rat)	Laser, 405 nm	100 mW	NR	10 min	NR	NR	Laser acupuncture at HT7 acupoint	Once daily for 14 days	Improved spatial memory in Morris water maze test; attenuated the decreased neuron density in CA3 and dentate gyrus, but not CA1 and CA2 regions; decreased activity of monoamine oxidase-B and acetylcholinesterase in the hippocampus; mitigated the decreased GSH-Px activity and the elevation of MDA level
Johnstone et al., (2014) [25]	Mouse Albino BALB/c: 50 mg/kg MPTP: (n _MPTP_ = 36) (n _MPTP + Transcranial PBM_ = 12) (n _MPTP + Remote PBM_ = 11) 75 mg/kg MPTP: (n _MPTP_ = 8) (n _MPTP + Transcranial PBM_ = 8) (n _MPTP + Remote PBM_ = 8) 100 mg/kg MPTP: (n _MPTP_ = 9) (n _MPTP + Transcranial PBM_ = 19) (n _MPTP + Remote PBM_ = 9)	Male 8 weeks old	MPTP: 50 mg/kg per mouse 75 mg/kg per mouse 100 mg/kg per mouse	LED, 670 nm	NR	50 mW/cm^2^ (at scalp)	90 s	4 J/cm^2^ (at scalp)	50 mg/kg MPTP: 8 J/cm^2^ (at scalp) 75 mg/kg MPTP: 12 J/cm^2^ (at scalp) 100 mg/kg MPTP: 16 J/cm^2^ (at scalp)	Transcranial irradiation to the head Remote irradiation to the dorsum	50 mg/kg MPTP: 2 irradiations over 2 days 75 mg/kg MPTP: 3 irradiations over 3 days 100 mg/kg MPTP: 4 irradiations over 4 days	In 50 but not 75 or 100 mg/kg MPTP doses: increased TH^+^ cell number in the SNc with both transcranial and remote irradiations
Moro et al., (2014) [25]	Mouse Albino BALB/c (n _Saline_ = 5) (n _Saline + Pulse PBM_ = 5) (n _Saline + Continuous PBM_ = 5) (n _MPTP_ = 5) (n _MPTP + Pulse PBM_ = 5) (n _MPTP + Continuous PBM_ = 5)	Male NR	MPTP: 50 mg/kg per mouse	LEDs, 670 nm	0.16 mW	Pulse irradiation 1.5 mW/cm^2^ Continuous irradiation 14.5 mW/cm^2^	Pulse irradiation: 90 s Continuous irradiation: 6 days continuously	Pulse irradiation: 0.13 J/cm^2^ Continuous irradiation: 7516.8 J/cm^2^	Pulse irradiation: 0.54 J/cm^2^ Continuous irradiation: 7516.8 J/cm^2^	Intracranial, implanted in the lateral ventricles	Pulse irradiation: 4 simultaneous irradiations over 30 h Continuous irradiation: 6 days continuously	For pulse irradiation group: significantly increased TH^+^ cell number in the SNc For continuous irradiation group: Non-significantly increased TH^+^ cell number in the SNc
Reinhart et al., (2015) [37]	Mouse Albino BALB/c (n _Saline_ = 11) (n _Saline + PBM_ = 11) (n _MPTP_ = 11) (n _MPTP + PBM_ = 11)	Male 8–10 weeks old	MPTP: 50 mg/kg per mouse	LEDs, 810 nm	0.16 mW	NR	90 s	14.4 mJ (at scalp)	57.6 mJ (at scalp)	Transcranial	4 simultaneous irradiations over 30 h	Improved locomotor activity at different time points including at immediately after first MPTP injection, at after sond PBM, at after fourth PBM, and 6 days after the last MPTP injection; increased TH^+^ cell number in the SNc
Darlot et al., (2015) [38]	Macaque monkey *Macaca fascicularis* (n _Control_ = 5) (n _MPTP (1.5 mg/kg)_ = 6) (n _MPTP (2.1 mg/kg)_ = 5) (n _MPTP (1.5 mg/kg) + PBM_ = 5) (n _MPTP (2.1 mg/kg) + PBM_ = 4)	Male 4–5 years old	MPTP: 1.5 mg/kg per monkey 2.1 mg/kg per monkey	Laser, 670 nm	10 mW	NR	MPTP (1.5 mg/kg) continuous irradiation (5 s ON/60 s OFF) for 5 days MPTP (2.1 mg/kg) continuous irradiation (5 s ON/60 s OFF) for 7 days	NA	MPTP (1.5 mg/kg): 25 J MPTP (2.1 mg/kg): 35 J	Intracranial, Implanted 1 to 2 mm to the left side of the midline in the midbrain	MPTP (1.5 mg/kg): continuous irradiation for 5 days MPTP (2.1 mg/kg): continuous irradiation for 7 days	For both irradiation groups: Improved clinical scores and behavioral activities as indicated by locomotive traces and distance moved and velocity as well as increased nigral dopaminergic cells For PBM (25 J) group: increased striatal TH^+^ terminals
Oueslati et al., (2015) [39]	AAV-Based Rat Genetic Model Sprague-Dawley (n _α-syn_ = 9) (n _α-syn + PBM (2.5 mW/cm_^2^_)_ = 7) (n _α-syn + PBM (5 mW/cm_^2^_)_ = 7)	Female NR	α-syn-induced toxicity: 2 μL of viral suspension per rat	Laser, 808 nm	NR	PBM (2.5 mW/cm^2^): 20.4 mW/cm^2^ (at scalp) or 2.5 mW/cm^2^ (at midbrain) PBM (5 mW/cm^2^): 40.8 mW/cm^2^ (at scalp) or 5 mW/cm^2^ (at midbrain)	100 s	PBM (2.5 mW/cm^2^): 4.08 J/cm^2^ (at scalp) or 0.50 J/cm^2^ (at midbrain) PBM (5 mW/cm^2^): 8.16 J/cm^2^ (at scalp) or 1 J/cm^2^ (at midbrain)	PBM (2.5 mW/cm^2^): 114.24 J/cm^2^ (at scalp) or 14 J/cm^2^ (at midbrain) PBM (5 mW/cm^2^): 228.48 J/cm^2^ (at scalp) or 28 J/cm^2^ (at midbrain)	Transcranial 2 irradiation spots of about 1 cm^2^ bilaterally on the head	All groups: once a day for 4 weeks	For both irradiation groups: decreased motor deficits (akinesia) as indicated by improvement of the use of the contralateral forepaw For PBM (5 mW/cm^2^) group: decreased nigral and striatal dopaminergic fiber loss
Moro et al., (2016) [40]	Macaque monkey *Macaca fascicularis* (n _Control_ = 3) (n _MPTP)_ = 5) (n _MPTP + PBM_ = 7)	Male 4–5 years old	MPTP: 1.8–2.1 mg/kg per monkey	Laser, 670 nm	10 mW	NR	Continuous irradiation (5 s ON/60 s OFF) for 25 days	NA	125 J	Intracranial, implanted in region 1–2 mm to the left hand side of the midline in the mid-brain	Continuous irradiation for 25 days	Improved clinical scores as indicated by locomotive traces; increased TH^+^ cell number in the SNc; no effect on the striatal TH^+^ terminal density
Salgado et al., (2016) [41]	Rat Albino Wistar (n _6OHDA_ = 20) (n _6OHDA + LED PBM_=20) (n _6OHDA + Laser PBM_ = 20)	NR NR	6OHDA bilateral microinjections of 15 μg per rat	LEDs, 627 nm Laser, 630 nm	LEDs: 70 mW Laser: 45 mW	LEDs: 70 mW/cm^2^ (at scalp) Laser: 45 mW/cm^2^ (at scalp)	LEDs: 57 s Laser: 88 s	LEDs: 4 J/cm^2^ (at scalp) Laser: 4 J/cm^2^ (at scalp)	LEDs: 28 J/cm^2^ (at scalp) Laser: 28 J/cm^2^ (at scalp)	Transcranial	All groups: once a day for 7 days	For laser and LEDs sources: increased locomotive traces in open field test; decreased TNF-α levels For LEDs source: increased IFN-γ levels For laser source: increased IL-2 levels no effect on the IL-4, IL-6 and IL-10 levels
Reinhart et al., (2016) [42]	Rat Wistar (n _Saline_ = 8) (n _6OHDA_ = 15) (n _6OHDA + Pulse PBM_ = 16) (n _6OHDA + Continuous PBM (0.16 mW)_ = 13) (n _6OHDA + Continuous PBM (333 nW)_ = 9)	Male 8 weeks old	6OHDA 7.5 μg/μL per rat	LEDs, 670 nm	Pulse irradiation: 0.16 mW Continuous irradiation (0.16 mW): 0.16 mW Continuous irradiation (333 nW): 333 nW	NR	Pulse irradiation: 90 s Continuous irradiation (0.16 mW): continuous irradiation for 23 days Continuous irradiation (333 nW): continuous irradiation for 23 days	NA	Pulse irradiation: 634 mJ Continuous irradiation (0.16 mW): 304 J Continuous irradiation (333 nW): 634 mJ	Intracranial, implanted in region near the SNc, incorporating the red nucleus and ventral tegmental area, toward the midline	Pulse irradiation: twice a day for 23 days Continuous irradiation (0.16 mW): continuous irradiation for 23 days Continuous irradiation (333 nW): continuous irradiation for 23 days	For pulse irradiation group: decreased rotational behavior at 21 days post-surgery; increased TH^+^ cell number in the SNc For continuous irradiation (0.16 mW) group: decreased rotational behavior at 14 and 21 days post-surgery; no effect on the TH^+^ cell number in the SNc For continuous irradiation (333 nW) group: no effect on the rotational behavior; no effect on the TH^+^ cell number in the SNc
Reinhart et al., (2016) [42]	Mouse Albino BALB/c (n _Saline_ = 9) (n _MPTP_ = 9) (n _MPTP + Pre-PBM_ = 9) (n _MPTP + Simultaneous PBM_ = 9) (n _MPTP + Post-PBM_ = 9) (n _MPTP + Pre- & Simultaneous PBM_ = 9) (n _MPTP + Post- & Simultaneous PBM_ = 9) (n _MPTP + Pre- & Post- & Simultaneous PBM_ = 9)	Male 8–10 weeks old	MPTP: 50 mg/kg per mouse	LEDs, 670 nm	NR	40 mW/cm^2^ (at scalp)	90 s	3.6 J/cm^2^ (at scalp)	Pre-PBM: 14.4 J/cm^2^ Simultaneous-PBM: 14.4 J/cm^2^ Post-PBM: 14.4 J/cm^2^ Pre- & Simultaneous PBM: 28.4 J/cm^2^ Post- & Simultaneous PBM: 28.4 J/cm^2^ Pre- & Post- & Simultaneous PBM: 43.2 J/cm^2^	Transcranial	Pre-PBM: twice a day for 2 days Simultaneous-PBM: twice a day for 2 days Post-PBM: twice a day for 2 days Pre- & Simultaneous PBM: twice a day for 4 days Post- & Simultaneous PBM: twice a day for 4 days Pre- & Post- & Simultaneous PBM: twice a day for 6 days	In all irradiation groups: increased locomotor activity in open field test by a similar magnitude and increased TH^+^ cell number in the SNc
El Massri et al., (2016) [43]	Macaque monkey *Macaca fascicularis* (n _Control_ = 5) (n _MPTP_) = 11) (n _MPTP + PBM_ = 6)	Male 4–5 years old	MPTP: 1.5–2.1 mg/k per monkey	Laser, 670 nm	10 mW	NR	Continuous irradiation (5 s ON/60 s OFF) for 5 or 7 days	NA	25 or 35 J	Intracranial, Implanted in 1 to 2 mm to the left side of the midline in the midbrain	Continuous irradiation for 5 or 7 days	Decreased number of GFAP^+^ astrocytes and astrocyte cell body size in the SNc and striatum; decreased microglia cell body size in the SNc and striatum
El Massri et al., (2016) [44]	Mouse *Albino BALB/c*: 2 days group (n _Saline_ = 7) (n _Saline + PBM_ = 10) (n _MPTP_ = 10) (n _MPTP+PBM_ = 10) 7 days group: (n _Saline_ = 7) (n _Saline + PBM_ = 10) (n _MPTP_ = 10) (n _MPTP+PBM_=10) 14 days group: (n _Saline_ = 7) (n _Saline + PBM_ = 10) (n _MPTP_ = 10) (n _MPTP + PBM (2 J/cm_^2^_)_ = 10) (n _MPTP + PBM (4 J/cm_^2^_)_ = 10)	Male 8–10 weeks old	MPTP: 50 or 100 mg/kg per mouse	LEDs, 670 nm	NR	40 mW/cm^2^ (at scalp)	90 s	4 J/cm^2^ (at scalp) or 0.5 J/cm^2^ (at brain)	2 days group: 8 J/cm^2^ (at scalp) or 1 J/cm^2^ (at brain) 7 days group: 8 J/cm^2^ (at scalp) or 1 J/cm^2^ (at brain) 14 days group (2 J/cm^2^): 16 J/cm^2^ (at scalp) or 2 J/cm^2^ (at brain) 14 days group (4 J/cm^2^): 32 J/cm^2^ (at scalp) or 4 J/cm^2^ (at brain)	Transcranial Holding probe at 1 cm from the head	2 days group: once a day for 2 days 7 days group: once a day for 2 days 14 days group (2 J/cm^2^): once a day for 4 days 14 days group (4 J/cm^2^): once a day for 8 days	In 7 days irradiation group: increased TH^+^ cell number in the SNc In 14 days (4 J/cm^2^ ) irradiation group: increased TH^+^ cell number in the SNc; decreased number of GFAP^+^ cells in the CPu
El Massri et al., (2017) [45]	Mouse *Albino BALB/c* (n _Saline_ = 5) (n _Saline + PBM_ = 3) (n _MPTP_ = 5) (n _MPTP + PBM_ = 4) Rat Wistar (n _Saline_ = 5) (n _6OHDA_ = 5) (n _6OHDA + PBM_ = 4) Macaque monkey *Macaca fascicularis* (n _Saline_ = 3) (n _Saline + PBM_ = 5) (n _MPTP_ = 5) (n _MPTP + PBM_ = 3)	Mouse: ~8 weeks old Rat: ~8 weeks old Monkey: 4–5 years old	Mouse: MPTP (50 mg/kg per mouse) Rat: 6OHDA (7.5 μg/μL) Monkey: MPTP (1.5 mg/kg per monkey)	Laser, 670 nm	Mouse: 0.16 mW Rat: 0.16 mW Monkey: 10 mW	NR	NR	NR	NR	Intracranial, Mouse: implanted in lateral ventricle Rat and Monkey: implanted in midline region of the midbrain	Mouse: Continuous irradiation for 30 h Rat: Continuous irradiation for 23 days Monkey: Continuous irradiation for 6 days	Mouse: no effect Rat: no effect Monkey: increased TH^+^ cell number and terminal density in the striatum; increased GDNF expression in the striatum
Reinhart et al., (2017) [46]	Mouse *Albino BALB/c* (n _Saline_ = 8) (n _MPTP_ = 8) (n _MPTP + 670 nm PBM_ = 8) (n _MPTP + 810 nm PBM_ = 8) (n _MPTP + Sequentially 670 & 810 nm PBM (15 mW)_ = 8) (n _MPTP + Sequentially 670 & 810 nm PBM (30 mW)_ = 8) (n _MPTP + Concurrently 670 & 810 nm PBM (15 mW)_ = 8) (n _MPTP + Concurrently 670 & 810 nm PBM (30 mW)_ = 8)	Male 8–10 weeks old	MPTP: 50 mg/kg per mouse	LED, 670 or 810 nm	15 or 30 mW	NR	45 or 90 s	2.7 J (at scalp)	670 nm PBM: 11 J 810 nm PBM: 11 J Sequentially 670 & 810 nm PBM (15 mW): 11 J Sequentially 670 & 810 nm PBM (30 mW): 22 J Concurrently 670 & 810 nm PBM (15 mW): 11 J Concurrently 670 & 810 nm PBM (30 mW): 22 J	Transcranial	All groups: twice a day for 2 days	In all irradiation groups: increased locomotor activity in open field test and increased TH^+^ cell number in the SNc Note: combination treatment groups exhibited a greater overall beneficial outcome
El Massri et al., (2018) [47]	Macaque monkey *Macaca fascicularis* (n _Control_ = 3) (n _Control + PBM_ = 3) (n _MPTP_ = 3) (n _MPTP + PBM_ = 3)	Male 4–5 years old	MPTP: 1.5 mg/kg per monkey	Laser, 670 nm	10 mW	NR	Continuous irradiation (5 s ON/60 s OFF) for 5 days	NA	25 J	Intracranial, Implanted in 1 to 2 mm to the left side of the midline in the midbrain	Continuous simultaneous irradiation for 5 days	No effect on the number and somal sizes of encephalopsin ^+^cells in the striatum
Kim et al., (2018) [48]	Mouse C57BL/6: (n _Saline_ = 10) (n _MPTP_ = 10) (n _MPTP + PBM_ = 10)	Male 10 weeks old	MPTP: 50 mg/kg per mouse	LED, 670 nm	NR	50 mW/cm^2^ (at skin)	180 s	9 J/cm^2^ (at skin)	18 J/cm^2^ (at skin)	Remotely; irradiation to the dorsum	Twice (24 h apart)	Increased TH^+^ cell number in the SNc; no effect on the density of TH^+^ terminations in the dorsal CPu
O’Brien and Austin (2019) [49]	Rat Sprague–Dawley (n _Vehicle_ = various) (n _Lipopolysaccharide_ = various) (n _Lipopolysaccharide + PBM_ = various)	Male NR	Lipopolysaccharide 10 μg per rat 20 μg per rat	LED, 675 nm	500 mW	40 mW/cm^2^ (at scalp)	88 s	3.6 J/cm^2^ (at scalp)	46.8 J/cm^2^ (at scalp)	Transcranial Holding probe at 1 cm from the head	Thirteen (once 2 h following the completion of the lipopolysaccharide injection + twice daily for 6 days)	With 10 µg lipopolysaccharide: increased TH^+^ cell number in the SNc; no effect on the IBA1^+^ cell densities in the SNc With 20 µg lipopolysaccharide: no significant effect on the motor behavior in the cylinder, rotarod and adjusted stepping tests
Miguel et al., (2019) [50]	Mouse C57BL/6: (n _Saline_ = 8) (n _MPTP_ = 6) (n _MPTP + PBM_ = 6)	Male 12 weeks old	MPTP: 80 mg/kg per mouse	LED, 675 nm	NR	50 mW/cm^2^ (at scalp)	180 s	9 J/cm^2^ (at scalp)	63 J/cm^2^ (at scalp)	Transcranial	Once a day for 7 days	Decreased vascular leakage in the SNc and CPu
Ganeshan et al., (2019) [51]	Mouse *Albino BALB/c* (n _Saline_ = 10) (n _MPTP_ = 10) (n _MPTP + PBM (2 days)_ = 10) (n _MPTP + PBM (5 days)_ = 10) (n _MPTP + PBM (10 days)_ = 10)	Male 10 weeks old	MPTP: 50 mg/kg per mouse	LED, 670 nm	NR	50 mW/cm^2^ (at skin)	90 s	4.5 J/cm^2^ (at skin)	PBM (2 days): 9 J/cm^2^ (at skin) PBM (5 days): 22.5 J/cm^2^ (at skin) PBM (10 days): 45 J/cm^2^ (at skin)	Remotely; irradiation to the dorsum and hind limbs	Once a day for 2, 5 or 10 days	In PBM (2 days) group: decreased Fos^+^ cell number in the CPu In PBM (5 days) group: decreased Fos^+^ cell number in the CPu In PBM (10 days) group: increased TH^+^ cell number in the SNc; decreased Fos^+^ cell number in the CPu; upregulated cell signaling and migration (including CXCR4^+^ stem cell and adipocytokine signaling), oxidative stress response pathways and modulated blood-brain barrier

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
