# Peer review of "Photobiomodulation for Parkinson’s Disease in Animal Models: A Systematic Review"

_biomolecules, 2020, doi:10.3390/biom10040610_

Round 1

Reviewer 1 Report

A major treatment approach for Parkinson's disease (PD) involves dopamine replacement therapy. Given that, long term dopamine replacement therapy can lead to other complications eg. dyskinesia, and here is no approved disease modifying therapies exist, alternative treatment strategies are unmet need. Here, Salehpour and Hamblin present a case in the form of "photobiomodulation - PBM". The authors provide a comprehensive review on PBM for PD in animal models. The idea being whether PBM can be a potential treatment strategy for PD in clinics. The manuscript is clearly written, and I find it suitable for a publication in Biomolecules. However, I have a few minor comments:

  1. Line 17: "PD causes movement disorders.....". This is an error. PD is one of the movement disorders. Please rephrase it.
  2. Line 33: "The dopamine producing neurons die off due to the accumulation of Lewy bodies...". This is heavily debated in the field. There is no consensus to make this claim.
  3. Line 36 & 37: "....among which mutation in GBA1 and SNCA are the most common". I would add LRRK2 in the list, perhaps start with LRKK2.
  4. Why most of the studies on the beneficial effects of PBM are carried out in toxin based animal models of PD (exception ref # 43 and 48)? Is there a strong connection between complex I inhibitors and PBM treatment? Can PBM be effective in other transgenic PD animal models?. Authors may discuss this. 

Author Response

  1. Line 17: "PD causes movement disorders.....". This is an error. PD is one of the movement disorders. Please rephrase it.

Author response: We have changed the sentence to “PD is a movement disorder and also shows various other symptoms affecting the brain and other organs.”

  1. Line 33: "The dopamine producing neurons die off due to the accumulation of Lewy bodies...". This is heavily debated in the field. There is no consensus to make this claim.

Author response: This point was also made by reviewer 2. We have changed the sentence to “The dopamine producing neurons die off, and one somewhat controversial theory to explain this is the accumulation of Lewy bodies containing aggregated α-synuclein inside the cells.”

  1. Line 36 & 37: "....among which mutation in GBA1 and SNCA are the most common". I would add LRRK2 in the list, perhaps start with LRKK2.

Author response: We have changed the sentence to “Only about 15% of PD patients are likely to have a genetic cause, among which mutations in leucine-rich repeat kinase 2 (LRRK2), GBA1 (glucocerebrosidase) and SNCA (α-synuclein) are the most common (3)”

  1. Why most of the studies on the beneficial effects of PBM are carried out in toxin based animal models of PD (exception ref # 43 and 48)? Is there a strong connection between complex I inhibitors and PBM treatment? Can PBM be effective in other transgenic PD animal models?. Authors may discuss this. 

Author response: We have added this sentence to the Discussion “It is expected that further animal studies will use PBM in genetically engineered models of PD rather than toxin-induced models, because these are now considered to be more representative of human disease “

Reviewer 2 Report

The manuscript by Salehpour & Hamblin details a systematic review of the effects of photobiomodulation in animal models of Parkinson’s disease. The findings collated and presented within the manuscript will provide a valuable resource for researchers in this field.

A few minor suggestions below:

  • I think it would be useful to have a paragraph in the Results section that provides a generalised summary of the results from the many animal studies that are reviewed. The take-home message seems to be that, across multiple animal models and using various PBM regimens, PBM is effective in mitigating dopaminergic cell loss from the nigrostriatal pathway and associated behavioural abnormalities.
  • It would be useful to include in Table 1 some information on the timing of treatment for the toxin models. For example, was PBM administered as a pre-, per- or post-conditioning intervention, or a combination of these?
  • Why is “Results” listed as a sub-section of “Materials and Methods” (2.4)?
  • In Table 1, there are two different studies that use citation 54 (El Massri 2017 and Reinhart 2017). Does this mean one reference is missing from the list? Please double-check references in Table 1.
  • Table 1 describes the outcome of Ganeshan et al 2019 as increased Fos+ cell number in the CPu; however, the cited paper reports decreased Fos+ cell number
  • Line 112: “six rodent studies were performed on rats, of which thirteen were on the Sprague-Dawley strain and one was on the albino Wistar strain” – these numbers don’t add up
  • Line 34: “neurons die off due to the accumulation of Lewy bodies” – this is contentious as, while Lewy bodies are present, it is still unclear whether they are the toxic species that kill neurons. I’d encourage the authors to soften this statement, e.g. “neurons die in association with the accumulation of Lewy bodies…”
  • There seems to be a formatting error throughout resulting in the “alpha” missing from “alpha-synuclein”

Author Response

  • It would be useful to include in Table 1 some information on the timing of treatment for the toxin models. For example, was PBM administered as a pre-, per- or post-conditioning intervention, or a combination of these?

Author response: This information is provided in the column headed “outcomes”

  •  
  • Why is “Results” listed as a sub-section of “Materials and Methods” (2.4)?

Author response: This has been corrected. It is now “3. Results”

  • In Table 1, there are two different studies that use citation 54 (El Massri 2017 and Reinhart 2017). Does this mean one reference is missing from the list? Please double-check references in Table 1.

Author response: Apologies for the errors in the references. These have been corrected.

  • Table 1 describes the outcome of Ganeshan et al 2019 as increased Fos+ cell number in the CPu; however, the cited paper reports decreased Fos+ cell number

Author response: Apologies for the error, it has been corrected

  • Line 112: “six rodent studies were performed on rats, of which thirteen were on the Sprague-Dawley strain and one was on the albino Wistar strain” – these numbers don’t add up

Author response: Apologies for the error, it has been corrected to “Also, six rodent studies were performed on rats, of which five were on the Sprague–Dawley strain and one was on the albino Wistar strain.

  • Line 34: “neurons die off due to the accumulation of Lewy bodies” – this is contentious as, while Lewy bodies are present, it is still unclear whether they are the toxic species that kill neurons. I’d encourage the authors to soften this statement, e.g. “neurons die in association with the accumulation of Lewy bodies…”

Author response: See response to Reviewer 1.

  • There seems to be a formatting error throughout resulting in the “alpha” missing from “alpha-synuclein”

Author response: This has been corrected.